# The Impact of Diagnostic Ureteroscopy Prior to Radical Nephroureterectomy on Oncological Outcomes in Patients with Upper Tract Urothelial Carcinoma: A Comprehensive Systematic Review and Meta-Analysis

**DOI:** 10.3390/jcm10184197

**Published:** 2021-09-16

**Authors:** Łukasz Nowak, Wojciech Krajewski, Joanna Chorbińska, Paweł Kiełb, Michał Sut, Marco Moschini, Jeremy Yuen-Chun Teoh, Keiichiro Mori, Francesco Del Giudice, Ekaterina Laukhtina, Chiara Lonati, Krzysztof Kaliszewski, Bartosz Małkiewicz, Tomasz Szydełko

**Affiliations:** 1University Center of Excellence in Urology, Department of Minimally Invasive and Robotic Urology, Wroclaw Medical University, 50-556 Wroclaw, Poland; joanna.chorbinska@gmail.com (J.C.); pk.kielb@gmail.com (P.K.); bartosz.malkiewicz@umed.wroc.pl (B.M.); tomasz.szydelko1@gmail.com (T.S.); 2Department of Urology, Ministry of Interior and Administration Hospital in Gdansk, 80-104 Gdansk, Poland; doktor.sut@gmail.com; 3Klinik für Urologie, Luzerner Kantonsspital, 6004 Lucerne, Switzerland; marco.moschini87@gmail.com; 4Department of Surgery, Prince of Wales Hospital, The Chinese University of Hong Kong, Hong Kong, China; jeremyteoh@surgery.cuhk.edu.hk; 5Comprehensive Cancer Center, Department of Urology, Medical University of Vienna, Vienna General Hospital, Währinger Gürtel 18-20, 1090 Vienna, Austria; morikeiichiro29@gmail.com (K.M.); katyalaukhtina@gmail.com (E.L.); 6Department of Urology, Jikei University School of Medicine, Tokyo 105-8461, Japan; 7Department of Maternal-Infant and Urological Sciences, Policlinico Umberto I Hospital, Sapienza University of Rome, Viale del Policlinico 155, 00161 Rome, Italy; francesco.delgiudice@uniroma1.it; 8Institute for Urology and Reproductive Health, Sechenov University, 119146 Moscow, Russia; 9Department of Urology, Spedali Civili of Brescia, 25123 Brescia, Italy; chiara.lonati@libero.it; 10Department of General, Minimally Invasive and Endocrine Surgery, Wroclaw Medical University, 50-556 Wroclaw, Poland; krzysztofkali@wp.pl

**Keywords:** upper tract urothelial carcinoma, radical nephroureterectomy, ureteroscopy, oncological outcomes

## Abstract

Background: The incidence of intravesical recurrence (IVR) following radical nephroureterectomy (RNU) is reported in up to 50% of patients with upper tract urothelial carcinoma (UTUC). It was suggested that preoperative diagnostic ureteroscopy (URS) could increase the IVR rate after RNU. However, the available data are often conflicting. Thus, in this systematic review and meta-analysis we sought to synthesize available data for the impact of pre-RNU URS for UTUC on IVR and other oncological outcomes. Materials and methods: A systematic literature search of the PubMed, Embase, and Cochrane Library databases was performed in June 2021. Cumulative analyses of hazard ratios (HRs) and their corresponding 95% confidence intervals (CI) were conducted. The primary endpoint was intravesical recurrence-free survival (IVRFS), with the secondary endpoints being cancer-specific survival (CSS), overall survival (OS), and metastasis-free survival (MFS). Results: Among a total of 5489 patients included in the sixteen selected papers, 2387 (43.4%) underwent diagnostic URS before RNU and 3102 (56.6%) did not. Pre-RNU diagnostic URS was significantly associated with worse IVRFS after RNU (HR = 1.44, 95% CI: 1.29–1.61, *p* < 0.001) than RNU alone. However, subgroup analysis including patients without biopsy during URS revealed no significant impact of diagnostic URS on IVRFS (HR = 1.28, 95% CI: 0.90–1.80, *p* = 0.16). The results of other analyses showed no significant differences in CSS (HR = 0.94, *p* = 0.63), OS (HR: 0.94, *p* = 0.56), and MFS (HR: 0.91, *p* = 0.37) between patients who underwent URS before RNU and those who did not. Conclusions: The results of this meta-analysis confirm that diagnostic URS prior to RNU is significantly associated with worse IVRFS, albeit with no concurrent impact on the other long-term survival outcomes. Our results indicate that URS has a negative impact on IVRFS only when combined with endoscopic biopsy. Future studies are warranted to assess the role of immediate postoperative intravesical chemotherapy in patients undergoing biopsy during URS for suspected UTUC.

## 1. Introduction

Upper tract urothelial carcinoma (UTUC) is a relatively rare neoplasm accounting for approximately 5–10% of all urothelial cancers [1]. According to the current European Association of Urology (EAU) guidelines, radical nephroureterectomy (RNU) with ipsilateral bladder cuff excision is a standard treatment for high-risk UTUC [2]. However, kidney-sparing surgeries (KSS), such as endoscopic ablation or segmental ureterectomy, are being increasingly used in clinical practice, particularly with low-risk disease, as they can preserve renal function and reduce morbidity without compromising oncological outcomes [2]. For this reason, appropriate preoperative risk stratification is of the utmost importance to allow adequate patient selection with respect to different therapeutic options. As risk stratification is predominantly based on tumour-related factors (e.g., tumour stage and grade), the utilization of preoperative predictive tools that yield reliable information regarding these factors is crucial in guiding the selection of candidates for conservative treatment [2].

Computerized tomography urography (CTU) is considered to be the most accurate imaging modality with which to diagnose UTUC. Combined with urine cytology evaluation, CTU may detect a significant proportion of UTUCs. According to current recommendations, diagnostic ureteroscopy (URS) should be performed if imaging and cytology are not sufficient for the diagnosis or risk stratification of the tumour [2]. However, several studies and previous meta-analyses have suggested that preoperative diagnostic URS could increase the IVR rate after RNU (reported in up to 50% of patients), which might be related to malignant urothelial cells backflow and tumour seeding during URS evaluation [3,4,5,6]. Nevertheless, the existing data are still conflicting. Furthermore, some technical aspects of URS, such as performance of biopsy during URS in relation to IVR, have not been closely evaluated to date.

In the following systematic review and meta-analysis with detailed exploratory analyses, we sought to comprehensively synthesize the available data regarding the impact of URS before RNU for UTUC on IVR, as well as other oncological outcomes.

## 2. Materials and Methods

The present systematic review and meta-analysis were performed according to the standard PRISMA (preferred reporting items for systematic reviews and meta-analysis) guidelines [7] and methods outlined in the Cochrane Handbook for Systematic Reviews of Interventions [8].

### 2.1. Search Strategy

Two review authors (Ł.N. and W.K.) independently performed a computerized systematic literature search of the PubMed, Embase, and Cochrane Library databases using a combination of the following terms/key words: (“upper tract urothelial carcinoma” OR “upper tract urothelial cancer” OR “upper urinary tract cancer” OR “upper tract urothelial neoplasm” OR “transitional cell carcinoma of the upper urinary tract” OR “UTUC” OR “UUTC”) AND (“ureteroscopy” OR “URS”). No specific time or language limitations were applied. The references of the relevant review articles were also manually screened to ensure that no additional eligible papers were inadvertently omitted. Additional screening was also performed on ahead-of-print articles published in the various urological journals. The last search was conducted on 30 June 2021.

### 2.2. Inclusion and Exclusion Criteria

Studies were evaluated for eligibility based on a predefined PICOS (population, intervention, comparison, outcome, and study design) approach. The inclusion criteria were as follows: (P) patients with UTUC treated with RNU; (I) diagnostic URS with or without biopsy prior to RNU (URS (+) group); (C) no diagnostic URS prior to RNU (URS (−) group); (O) primary outcome was intravesical recurrence-free survival (IVRFS), and secondary outcomes were cancer-specific survival (CSS), overall survival (OS), and metastases-free survival (MFS); (S) prospective and retrospective studies.

The exclusion criteria were as follows: (1) studies were meeting abstracts, review papers, case reports, letters, and editorials; (2) studies reported no sufficient data to estimate the hazard ratios (HRs) and 95% confidence intervals (CIs); (3) studies included patients who underwent tumour ablation during URS; (4) studies included patients who underwent kidney-sparing surgery.

### 2.3. Data Extraction

The data extraction process was completed independently by two review authors (Ł.N. and W.K.). Following initial screening of results using the titles and abstracts, full text screening and study selection was carried out using a standardized item form. Disagreements or discrepancies were resolved by discussion with a third author who was not involved in the initial screening process (T.S.). Study-related data and clinicopathological characteristics of included articles were initially extracted. Subsequently, the outcome measurements of IVRFS, CSS, OS, MFS (HRs and 95% CIs) were extracted. Missing information or clarifications were sought by contacting the primary authors. However, no additional data were retrieved.

### 2.4. Methodological Quality and Risk of Bias Assessment

All selected, nonrandomized studies were assessed for their methodological quality using the Newcastle–Ottawa scale (NOS), with the methodological quality stratified by score as: low (0–3), moderate (4–6), or high (7–9) [9]. The “risk of bias” (RoB) for each included manuscript was assessed according to the principles outlined in the *Cochrane Handbook for Systematic Reviews of Interventions*. The articles were reviewed based on their adjustment for major confounders: age, gender, tumour location, tumour multifocality, pathological tumour stage, pathological tumour grade, presence of concomitant carcinoma in situ (CIS). The risk of confounding bias was considered to be high if the confounder was not controlled for in multivariate analysis. Any disagreements and discrepancies were resolved by consensus or recourse to the third author (T.S.).

### 2.5. Statistical Analysis

The effect measures for the outcomes of survival (IVRFS, CSS, OS, and MFS) were HRs and 95% CIs, which were primarily extracted from the included articles. For publications that did not present HRs and 95% CIs, methods reported by Tirney et al. were used to incorporate summary time-to-event data into the meta-analysis [10]. The statistical significance of the pooled HRs was evaluated by the Z test. Statistical pooling of the effect measures was based on the level of heterogeneity among the studies. Significant heterogeneity was indicated by either a ratio of >50% in I^2^ statistics or a *p*-value of ≤0.05 in Cochran’s Q test, which led to the use of the random effects model. When no significant heterogeneity was observed, a fixed effects model was used for calculations. For each comparison, we conducted sensitivity analysis and publication bias assessment (based on the visual interpretation of funnel plots and the results of Egger’s test). For all tests, a *p*-value ≤ 0.05 was considered to be a statistically significant difference.

## 3. Results

### 3.1. Literature Selection and Baseline Characteristics of Included Studies

The detailed flow diagram of study selection with subsequent exclusions is presented in Figure 1. Eventually, sixteen retrospective studies were included in this systematic review and meta-analysis [11,12,13,14,15,16,17,18,19,20,21,22,23,24,25,26]. Among a total of 5489 patients in the selected papers, 2387 (43.4%) underwent diagnostic URS before RNU (URS (+) group) and 3102 (56.6%) did not (URS (−) group). The baseline characteristics of the eligible manuscripts are presented in Table 1. Geographically, nine studies were conducted in Asia [13,15,17,18,19,20,21,25,26], four in North America [12,14,23,24], and three in Europe [11,16,22]. Included articles reported cases from the years 1985 to 2019. The median follow-up periods for whole or individual cohorts ranged from 21.4 months to 76.8 months, and a pooled follow-up was 42.1 months. Among sixteen selected papers, thirteen provided data regarding IVRFS [11,13,15,16,17,18,19,20,21,23,24,25,26] whilst seven [13,15,17,20,21,22,23], seven [12,13,14,17,21,23,24] and five [14,17,20,22,23] studies reported CSS, OS, and MFS, respectively.

The NOS score for the included studies ranged from 6 to 9, with an overall mean score of 7.8. The methodological quality of the eligible manuscripts was therefore moderate or high (Table 1), which was considered appropriate for this systematic review and meta-analysis. All selected papers carried a high RoB, which was primarily related to their retrospective design. An assessment of confounding factors used for adjustments in Cox proportional hazard regression models for each study is presented in Figure 2.

### 3.2. Clinicopathological Characteristics of Included Studies

The clinicopathological characteristics of patients in selected articles are presented in Table 2. The median or mean age of patients in included studies ranged from 63.8 to 73 years, with no statistically significant differences observed between URS (+) and URS (−) groups. The male predominance was reported in twelve [11,12,13,14,15,16,18,22,23,24,25,26] out of sixteen articles. Six studies [11,13,16,18,19,23] presented populations without a history of bladder cancer prior to RNU. The proportion of patients with a history of bladder cancer in ten remaining articles [12,14,15,17,20,21,22,24,25,26] ranged from 4.5%–42.7% and 0%–44% in URS (+) and URS (−) groups, respectively. In four [12,17,21,26] out of sixteen articles, all patients underwent URS with biopsy, whilst in eight articles [11,15,16,18,20,22,24,25] biopsy rates during URS ranged from 59.5% to 92.6%. Four publications [13,14,19,23] lacked data regarding performance of URS biopsy. Two studies [24,25] distinguished specific subpopulations based on the performance of the biopsy during URS, and analyzed differences in IVRFS with reference to patients who did not undergo URS before RNU. Only three studies provided information on whether or not a selective cytology was performed in the URS (+) group, and none presented the detailed results of that examination [11,15,22]. In the majority of eligible publications, URS (+) and URS (−) groups were well-matched in terms of pathological tumour characteristics. Only four [11,19,21,22] out of sixteen studies reported significant differences in pathological tumour stage between patients who underwent URS and those who did not (with a higher proportion of ≤pT2 tumours in the URS (+) group). In one study [22], a lower proportion of G3 tumours was observed in the URS (+) group. Some trials reported significantly higher rates of ureteral tumours in patients who had undergone URS before RNU [13,16,18,19,20,22,23]. If reported, LNI and concomitant CIS rates were similar in most studies. The proportion of patients receiving intravesical installation after RNU ranged from 9.7% to 16% and 1.9% to 22% in URS (+) and URS (−) groups, respectively, based on data collected from two articles [11,24]. Data regarding the administration of perioperative systemic chemotherapy were presented in three papers [16,24,25], whereas in other articles patients receiving neoadjuvant or adjuvant systemic chemotherapy were initially excluded, or no data were available. Additional information regarding the technical details of URS and RNU are presented in Appendix A.

Data regarding the characteristics of intravesical recurrences are provided in the Appendix A. Rates of IVR reported in included studies ranged from 27.0% to 59.0% and 16.7% to 27.8% in the URS (+) and URS (−) groups, respectively. Only one study [11] provided the pathological characteristics of bladder recurrences, reporting that all intravesical recurrences were <T2 stage, and that the majority of tumours were low-grade.

#### 3.3.1. Main Analyses

Data for IVRFS were extractable from 13 studies [11,13,15,16,17,18,19,20,21,23,24,25,26]. One study [18] presented HRs as stratified by the timing of RNU after URS (RNU at the same day or later) and one study [24] presented HRs as stratified by the performance of the endoscopic biopsy during URS. Estimates in these studies were combined using a random effects model and subsequently used in primary IVRFS analysis. The pooled results indicated that diagnostic URS prior to RNU was significantly associated with worse IVRFS after RNU compared to RNU alone (HR = 1.44, 95% CI: 1.29–1.61, *p* < 0.001) (Figure 3). Heterogeneity between studies was considered nonsignificant with an I^2^ = 37% (*p* = 0.09), thus, a fixed effects model was used for data synthesis.

Data for CSS were extractable from seven studies [13,15,17,20,21,22,23]. The pooled results indicated that compared to RNU alone, diagnostic URS prior to RNU was not significantly associated with worse CSS after RNU than RNU alone (HR = 0.94, 95% CI: 0.75–1.19, *p* = 0.63) (Figure 4). Heterogeneity between studies was considered nonsignificant with an I^2^ = 29% (*p* = 0.21), thus, a fixed effects model was used for data synthesis.

Data for OS was extractable from seven studies [12,13,14,17,21,23,24]. The pooled results indicated that compared to RNU alone, diagnostic URS prior to RNU was not significantly associated with worse OS after RNU than RNU alone (HR = 0.94, 95% CI: 0.75–1.17, *p* = 0.56) (Figure 5). Heterogeneity between studies was considered nonsignificant with an I^2^ = 47% (*p* = 0.08), thus, a fixed effects model was used for data synthesis.

Data for MFS was extractable from five studies [14,17,20,22,23]. The pooled results indicated that compared to RNU alone, diagnostic URS prior to RNU was not significantly associated with worse MFS after RNU than RNU alone (HR = 0.91, 95% CI: 0.74–1.12, *p* = 0.37) (Figure 6). Heterogeneity between studies was considered nonsignificant with an I^2^ = 0% (*p* = 0.42), thus, a fixed effects model was used for data synthesis.

An examination of the funnel plots combined with an analysis of the Egger’s test results did not demonstrate a significant publication bias (Figure 7). In sensitivity analyses omitting enrolled studies in turn, the results showed that the pooled HRs did not differ significantly, suggesting that the findings of the primary analyses were stable (Appendix A).

#### 3.3.2. Exploratory Subgroup Analyses

Following assessment of primary analyses, prespecified exploratory subgroup analyses of IVRFS were performed (Table 3). Given that CSS, OS, and MFS were reported in very few cohorts, no subgroup analyses were conducted for these oncological outcomes.

Subgroup analysis including patients without biopsy during URS revealed no significant impact of URS on IVRFS (HR = 1.28, 95% CI: 0.90–1.80, *p* = 0.16) with reference to patients who did not undergo URS before RNU. When patients receiving a biopsy during URS were separately analyzed, results were similar to primary IVRFS analysis (HR = 1.38, 95% CI: 1.20–1.60, *p* < 0.001). Also, no association between URS and worse IVRFS was found in the subset of patients with a history of bladder tumours (HR = 1.71, 95% CI: 1.48–1.97, *p* < 0.001). In all other subgroup analyses, URS before RNU was significantly associated with worse IVRFS compared to RNU alone. Similar outcomes were demonstrated in the primary IVRFS analysis.

## 4. Discussion

In the present systematic review and meta-analysis, we attempted to provide a comprehensive summary of the evidence available to assess the impact of diagnostic URS prior to RNU on oncological outcomes in patients with UTUC. Our primary analyses confirmed that diagnostic URS before RNU had a significant negative impact on IVRFS after RNU without compromising the other oncological outcomes (CSS, OS, and MFS).

Kulp et al. and Lim et al. initially explored the potential role of URS in cancer dissemination [27,28]. The former reported on a series of 13 patients who underwent URS followed by RNU, demonstrating the absence of vascular or lymphatic space tumour cells in their surgical specimens [27], while the latter described a case of suspected lymphatic invasion attributable to high intrarenal pressures during URS [28]. Subsequently, several theories, mainly based on intraluminal tumour seeding or intraepithelial cancer migration hypotheses, have been advanced to explain the observed higher rates of IVR after RNU preceded by URS [29,30]. Audenet et al. showed that the majority of bladder tumours following RNU are clonally related, supporting the hypothesis that IVR occurs through neoplastic cell implantation, rather than representing a second primary tumour [31]. Recent molecular studies demonstrated that UTUC and bladder urothelial carcinoma share mutations in similar genes, but at varying frequencies, which recapitulate with their metachronous recurrences [32]. Such differences in quantitative gene alterations might contribute to the behaviour of these two cancers and imply that they and their metachronous recurrences should be treated as two related yet distinct entities [32].

To the best of our knowledge, this is the first meta-analysis exploring the risk of IVR associated with URS combined with endoscopic biopsy prior to RNU. Although several studies and previous meta-analyses demonstrated that URS combined with tumour biopsy had no influence on IVR when compared to URS alone (only tumour visualization) [5,6,11,15], the direct comparisons of these subgroups (URS biopsy and URS alone) with reference to patients who did not undergo URS before RNU were not found. Herein, in this study, we found that patients not undergoing tumour biopsy during URS had comparable IVRFS to those diagnosed with UTUC based on imaging studies alone. Conversely, those undergoing URS biopsy were significantly associated with worse IVRFS than those not undergoing URS. These findings indicate that increased IVR risk after URS followed by RNU is related mainly to direct tumour manipulation during endoscopic biopsy. Our results thus provide the rationale for immediate administration of intravesical chemotherapy after URS only in those undergoing endoscopic biopsy.

Another potential risk associated with URS being performed prior to RNU raised by several authors is a delay in providing definitive treatment, which could be associated with poorer survival outcomes [33,34]. URS before RNU could significantly prolong the time between presentation and definitive treatment, especially when the histopathological confirmation is expected [33,34]. A recent multicentre cohort study conducted by Lee et al. showed that delaying RNU for more than three months was associated with poor overall survival, thus, a delay between URS and RNU should not exceed this period [35]. In our meta-analysis we demonstrated that performing URS prior to RNU does not affect long-term oncological outcomes, such as CSS and OS, which is consistent with previous reports [5,6]. It could be explained by the fact that reported delay between URS and RNU was not longer than two months in the majority of included studies.

Unfortunately, certain technical aspects of URS, such as the duration of the procedure or irrigation pressure, have not been rigorously explored in the included studies. Moreover, only a minority of studies reported on the use of homogenous URS techniques [11,20]. Results of these studies suggest that both rigid and flexible URS significantly increase the risk of IVR following RNU. Additionally, in a study conducted by Baboudjian et al., the risk of IVR was not decreased even with the implementation of technical precautions to avoid contact of the bladder mucosa with contaminated urine from the upper urinary tract (use of ureteral sheath, or drainage with mono-J and bladder catheter) [11]. Moreover, we were unable to synthetize data pertaining to any urine cytology results. Although some recent studies demonstrated that barbotage cytology could detect up to 91% of UTUC (with a similar accuracy to biopsy histology) [36], we were not able to draw any conclusions over whether or not it could be used as a standalone procedure (instead of endoscopic biopsy) without compromising IVRFS.

As tumour location, bladder cancer history, and bladder cuff management are considered to be important factors associated with IVR after RNU [2,3,37], we investigated the impact of URS followed by RNU on IVRFS in specific patient cohorts in our exploratory subgroup analyses. However, only one study stratified risk estimates by tumour location and reported that URS biopsy before RNU increased the risk of IVR in patients with renal pelvic tumours, but this was not associated with IVR in patients with ureteral tumours [26]. Study results suggest that the indications for URS combined with endoscopic biopsy should vary according to tumour location. If a ureteral tumour is suspected in other preoperative examinations, URS biopsy should not be spared to allow for pathologic confirmation of the tumour before definitive treatment. Furthermore, even after excluding patients with a previous history of bladder cancer, patients undergoing URS were shown to be still at higher risk of IVR than those not undergoing URS. Thus, of the studies included, we subsequently confined our analysis to those involving bladder cuff management and found significantly worse IVRFS in those undergoing URS, a similar finding to that of the primary analysis.

Current guidelines recommend a single postoperative instillation of intravesical chemotherapy after RNU, as several trials and meta-analyses showed an approximately 40% relative risk reduction of IVR after administration of a single postoperative dose of intravesical chemotherapy (mitomycin C or pirarubicin) [2,38]. Although we demonstrated that URS prior to RNU is associated with significantly worse IVRFS than RNU alone in patients not receiving immediate intravesical installation following RNU, we were unable to perform subgroup analyses for the subset of patients who received adjuvant intravesical treatment due to a paucity of data. Moreover, there is currently an intense discussion over whether or not bladder chemoprophylaxis should be administered immediately after URS for the diagnosis or treatment of UTUC [39]. At present, there are no recommendations for such a procedure, with only indirect data available. Based on the several studies demonstrating the reduced risk of bladder recurrences following transurethral resection of bladder urothelial carcinoma and immediate single instillation of intravesical chemotherapy [40], as well as results of the Olympus trial confirming the effects of mitomycin C gel for ablative UTUC treatment [41], there is a considerable assumption that the single installation of a chemotherapeutic agent after diagnostic URS could significantly decrease the IVR rate.

Unfortunately, we were unable to perform subgroup analyses for populations of patients who received any form of neoadjuvant or adjuvant treatment due to a lack of data. Given that systemic neoadjuvant (NAC) and adjuvant (AC) chemotherapy has a significant beneficial effect on oncological outcomes in patients with UTUC [42], further studies including cohorts of patients receiving perioperative chemotherapy are necessary to allow for preliminary conclusions regarding the impact of URS prior to RNU on IVRFS in such settings.

Based on the results of this systematic review and meta-analysis, performance of URS combined with endoscopic biopsy should be considered only in cases that pose a challenging diagnostic dilemma (e.g., nonspecific CTU findings or equivocal urine cytology), as it significantly increases the IVR risk. Obtaining a definitive pathological diagnosis before RNU should not be a strong enough indication for URS biopsy when NAC is not planned and the patient can be assigned as high risk based on other factors, such as multifocality or size of the lesion from the CTU or positive cytology results. However, as NAC was demonstrated to have potential survival benefit in UTUC patients [42], and most oncologists would not begin neoadjuvant chemotherapy without histopathological confirmation, URS biopsy may continue to play an essential role in patients scheduled to NAC, even in cases in which preoperative imaging and cytology is highly suggestive of UTUC. Moreover, performance of URS combined with endoscopic biopsy should be always discussed as a potential diagnostic modality in each individual case when the patient might benefit from KSS.

Although the influence of performing URS before RNU has been previously evaluated by other groups [4,5,6], our current study has several strengths. First, we included the latest available studies based on the latest literature search up to 30 June 2021. Second, we did not exclude non-English articles, in contrast to the previous meta-analyses. Third, our meta-analysis included the most detailed exploratory subgroup analyses performed to date. Nonetheless, despite its strengths, this review is not devoid of limitations. First, the strength of the conclusions that can be drawn from our meta-analysis is still limited, given the fact that all included studies were retrospective with their own unavoidable limitations, such as selection bias. Second, the adjustments for confounders in the Cox regression analyses were not uniform in the included trials and some studies provided only univariable data, which might introduce additional bias. Third, additional data regarding factors such as URS technique or performance of urine cytology during URS were scarcely or not uniformly reported, and the influence of such significant heterogeneity could not be fully excluded. Fourth, the results of subgroup analyses should be interpreted carefully, as some of them were based on a limited number of patients.

## 5. Conclusions

The results of this systematic review and meta-analysis confirm that diagnostic URS prior to RNU is significantly associated with worse IVRFS, albeit with no concurrent impact on the other long-term survival outcomes. Our results indicate that URS has a negative impact on IVRFS only when combined with endoscopic biopsy. Future studies are needed to assess the role of immediate postoperative intravesical chemotherapy in patients undergoing biopsy during URS for suspected UTUC.

## Figures and Tables

**Figure 1 jcm-10-04197-f001:**
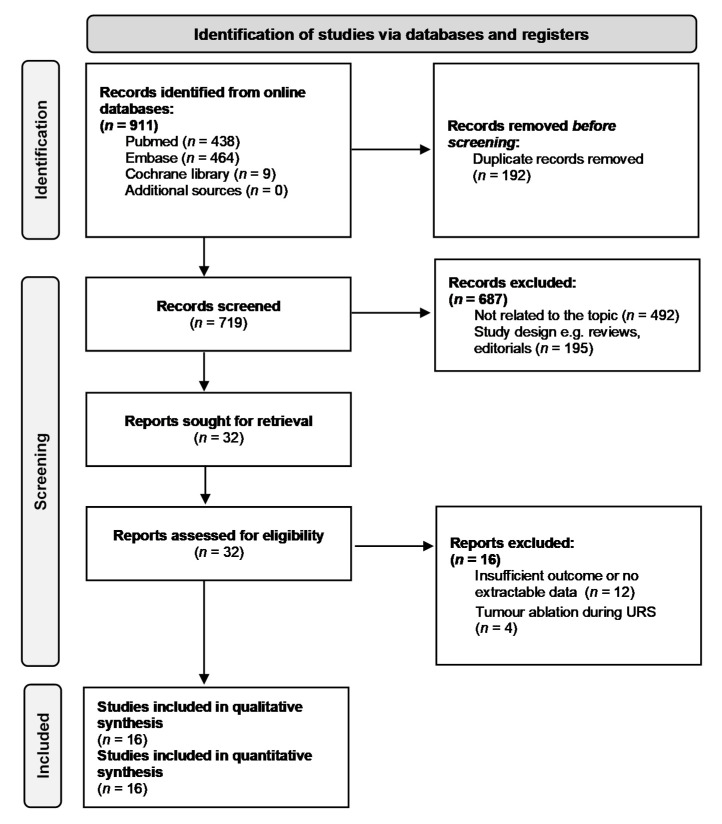
Flow diagram of study selection: The literature search yielded a total of 911 articles. All citations were exported to the reference manager and duplicate references (*n* = 192) were removed. After screening of the titles and abstracts, 687 papers were excluded due to inappropriate article type (*n* = 195) or irrelevance to the present topic (*n* = 492). Out of 32 reports assessed for eligibility, 12 and 4 articles were excluded for insufficient outcome and inclusion of patients who received tumour ablation during URS, respectively. Finally, sixteen studies were included. URS = ureteroscopy.

**Figure 2 jcm-10-04197-f002:**
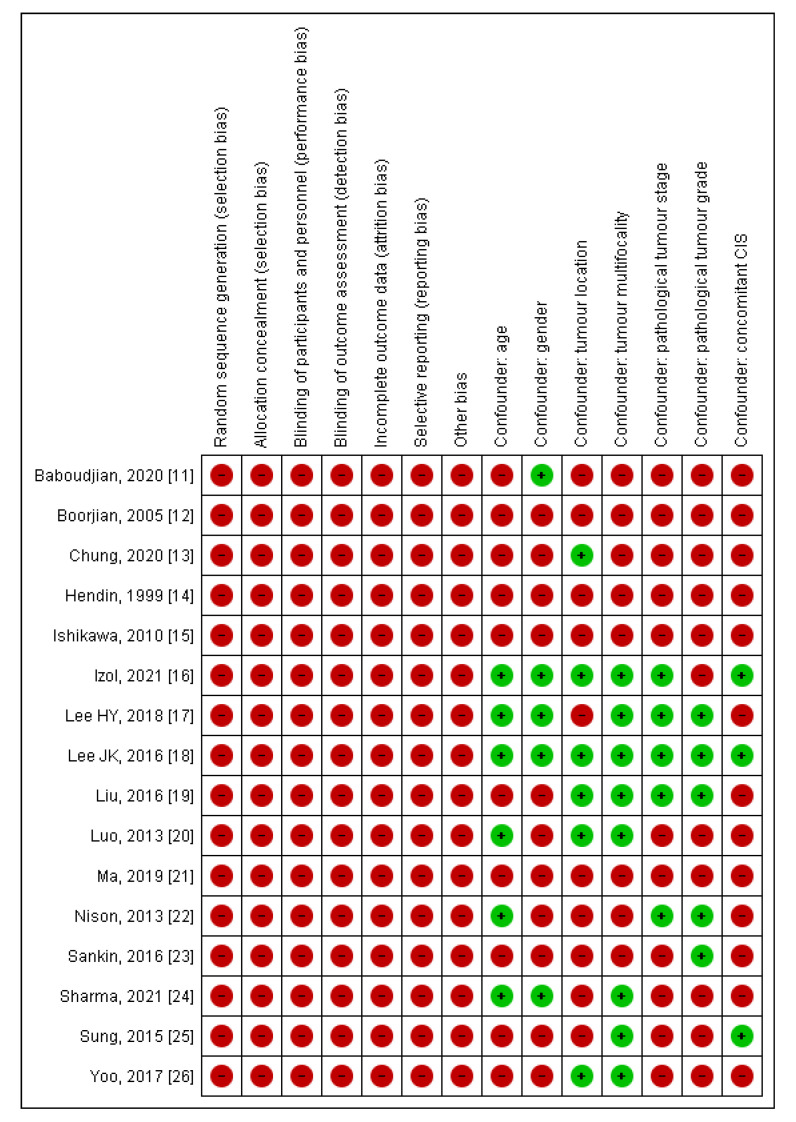
The risk of bias and confounding assessment for all included studies [11,12,13,14,15,16,17,18,19,20,21,22,23,24,25,26]. The green circle represents a low risk of bias and confounfing. The red circle represents a high risk of bias and confounding. CIS = carcinoma in situ.

**Figure 3 jcm-10-04197-f003:**
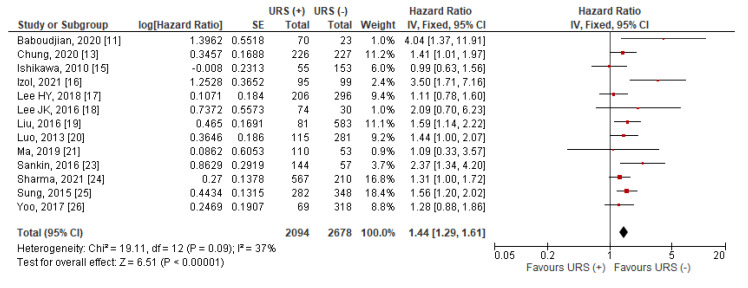
Forest plot and meta-analysis of intravesical recurrence-free surical (IVRFS) [11,13,15,16,17,18,19,20,21,22,23,24,25,26]. CI = confidence interval; IV = inverse variance; SE = standard error; URS = ureteroscopy.

**Figure 4 jcm-10-04197-f004:**
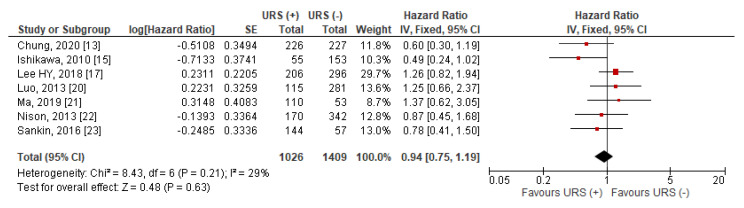
Forest plot and meta-analysis of cancer-specific survival (CSS) [13,15,17,20,21,22,23]. CI = confidence interval; IV = inverse variance; SE = standard error; URS = ureteroscopy.

**Figure 5 jcm-10-04197-f005:**
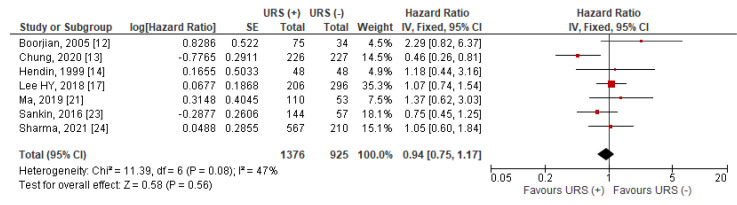
Forest plot and meta-analysis of overall survival (OS) [12,13,14,17,21,23,24]. CI = confidence interval; IV = inverse variance; SE = standard error; URS = ureteroscopy.

**Figure 6 jcm-10-04197-f006:**
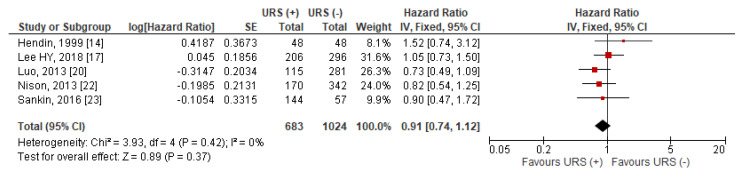
Forest plot and meta-analysis of metastasis-free survival (MFS) [14,17,20,22,23]. CI = confidence interval; IV = inverse variance; SE = standard error; URS = ureteroscopy.

**Figure 7 jcm-10-04197-f007:**
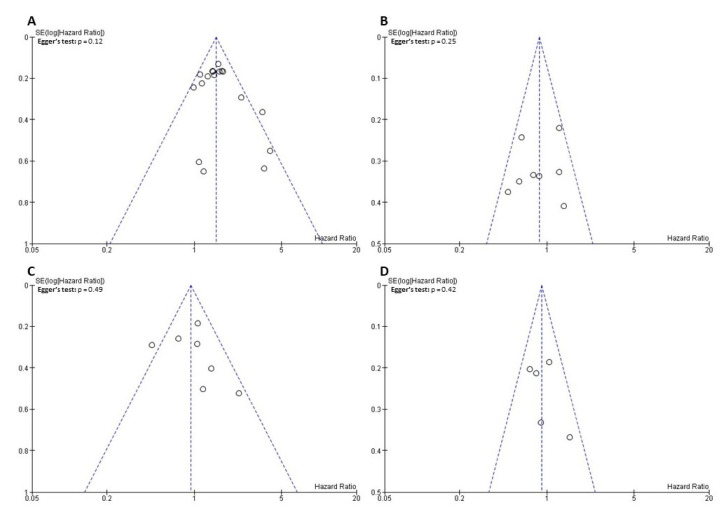
Funnel plot for the evaluation of potential publication bias: (**A**) intravesical recurrence-free survival; (**B**) cancer-specific survival; (**C)** overall survival; (**D**) metastasis-free survival. SE = standard error.

**Table 1 jcm-10-04197-t001:** Baseline characteristics and quality assessment of included studies.

First Author, Year [Reference]	Country	Study Design	Recruitment Period, Years	No. of Patients, *n* URS (+)/URS (−)	Follow-Up, Months URS (+)/URS (−)	Reported Outcomes of Interest	Methodological Quality (NOS)
Baboudjian, 2020 [11]	France	R, single-centre	2005–2017	70/23	Median: 35/34	IVRFS	8
Boorjian, 2005 [12]	United states	R, single-centre	1993–2003	75/34	Mean: 40.1/38.7	OS	6
Chung, 2020 [13]	Korea	R, single-centre	2003–2018	226/227	Median: 46.1/36.9	IVRFS, CSS, OS	7
Hendin, 1999 [14]	United States	R, single-centre	1985–1995	48/48	Mean: 50.4/42.4	OS, MFS	8
Ishikawa, 2010 [15]	Japan	R, multi-centre	1990–2005	55/153	Median: 35/51	IVRFS, CSS	7
Izol, 2021 [16]	Turkey	R, multi-centre	2005–2019	95/99	Mean: 36.4/41.8	IVRFS	8
Lee HY, 2018 [17]	Taiwan	R, single-centre	1990–2013	206/296	Mean: 76.8	IVRFS, CSS, OS, MFS	9
Lee JK, 2016 [18]	Korea	R, single-centre	2003–2012	74 */30	Mean: 34.4	IVRFS	8
Liu, 2016 [19]	China	R, single-centre	2000–2011	81/583	Median: 48	IVRFS	9
Luo, 2013 [20]	Taiwan	R, single-centre	2004–2010	115/281	Mean: 42.4/38.9	IVRFS, CSS, MFS	9
Ma, 2019 [21]	China	R, single-centre	2007–2016	110/53	Median: 40	IVRFS, CSS, OS	7
Nison, 2013 [22]	France	R, multi-centre	1995–2011	170/342	Median: 21.4/24	CSS, MFS	7
Sankin, 2016 [23]	United States	R, single-centre	1994–2012	144/57	Median: 64.8	IVRFS, CSS, OS, MFS	7
Sharma, 2021 [24]	United States	R, single-centre	1995–2019	567 **/210	Median 29.3	IVRFS, OS	9
Sung, 2015 [25]	Korea	R, single-centre	1994–2013	282/348	Median: 30.1/39.3	IVRFS	8
Yoo, 2017 [26]	Korea	R, single-centre	1998–2012	69/318	Median: 62	IVRFS	7

***** number of patients who underwent URS regardless of following RNU time; ** number of patients who underwent URS regardless of biopsy status. Abbreviations: CSS = cancer-specific survival; IVRFS = intravesical recurrence-free survival; MFS = metastasis-free survival; NOS = Newcastle–Ottawa scale; OS = overall survival; R = retrospective; RNU = radical nephroureterectomy; URS = ureteroscopy.

**Table 2 jcm-10-04197-t002:** Clinical and pathological characteristics of included studies.

First Author, Year [Reference]	Age URS (+)/URS (−)	Male Gender, % URS (+)/URS (−)	History of BC, % URS (+)/URS (−)	URS Biopsy, % URS (+)/URS (−)	Ureteral Location, % URS (+)/URS (−)	Pathological Stage ≥pT3, % URS (+)/URS (−)	Pathological Grade (HG or G3), % URS (+)/URS (−)	Concomitant CIS, % URS (+)/URS (−)	LNI, % URS (+)/URS (−)	Intravesical Installation Post RNU, % URS (+)/URS (−)	NAC, % (URS (+)/URS (−)	AC, % URS (+)/URS (−)
Baboudjian, 2020 [11]	72/73 ^a^	71.4/73.9	Excluded	88.6	42.9/30.4	32.9/60.9 *	HG: 72.9/87.0	10.0/0	3.0/9.0	16.0/22.0	Excluded	Excluded
Boorjian, 2005 [12]	68.4/67.1 ^b^	70.7/70.6	42.7/44.1	100.0	49.3/47.1	21.4/17.6	HG: 28.0/20.6	NR	NR	0/0	NR	NR
Chung, 2020 [13]	65.9/67.0 ^b^	68.6/72.7	Excluded	NR	57.1/31.7 *	35.4/38.8	G3: 45.1/54.2	NR	9.7/10.1	NR	NR	NR
Hendin, 1999 [14]	63.2/67.5 ^b^	77.1/64.6	29.2/35.4	NR	47.9/29.2	19.4/29.8	G3: 29.0/47.9	NR	NR	NR	NR	NR
Ishikawa, 2010 [15]	71/70 ^a^	65.5/67.3	23.6/17.0	65.5	50.9/45.1	21.8/34.0	HG: 21.8/35.3	NR	1.8/8.5 *	NR	Excluded	Excluded
Izol, 2021 [16]	66.3 ^b^	77.9/80.8	Excluded	61.1	36.8/17.2 *	49.5/54.5	HG: 71.6/66.7	8.4/6.1	6.4/9.1	Excluded	8.4/3.0	12.6/19.2
Lee HY, 2018 [17]	66.1/65.7 ^b^	41.3/45.6	29.6/29.4	100.0	46.1/42.6	33.5/35.2	HG: 76.7/78.7	NR	7.8/7.8	0/0	NR	NR
Lee JK, 2016 [18]	67.3/67.5 ^b^	64.9/76.6	0/0	59.5	8.1/30.0 *	13.5/23.3	HG: 75.7/76.7	17.6/10.0	NR	NR	0/0	0/0
Liu, 2016 [19] #	65.9/66.6 ^b^	38.3/45.3	Excluded	NR	67.9/41.3 *	19.8/32.6 *	HG: 37.0/43.3	4.9/2.6	3.7/7.5	NR	0/0	0/0
Luo, 2013 [20]	65.9/66.6 ^b^	47.0/48.4	27.0/27.4	73.9	72.2/53.0 *	21.7/29.2	HG: 88.7/90.7	33.9/37.4	Excluded	0/0	0/0	NR
Ma, 2019 [21]	66.1/69.0	46.4/37.8	4.5/0	100.0	53.6/39.6	23.6/39.6 *	HG: 70.0/75.5	NR	2.7/3.8	NR	NR	NR
Nison, 2013 [22]	68.8/70.1 ^a^	65.9/69.0	22.9/21.4	66.8	41.8/29.5 *	34.1/43.6 *	G3: 50.6/61.4 *	NR	4.1/9.4	NR	Excluded	NR
Sankin, 2016 [23]	70/71 ^a^	60.4/40.0 *	0/0	NR	28.0/12.0 *	27.1/36.8	HG: 78.0/86.0	NR	NR	NR	NR	NR
Sharma, 2021 [24] ##	72.8/72.5 ^a^	65.1/69.0	31.2/34.3	78.0	NA	≥pT2: 38.8/45.7	HG: 59.8/59.1	16.6/17.6	NR	7.9/1.9 *	8.3/6.7	13.2/21.3
Sung, 2015 [25]	64/65 ^a^	75.2/72.7	15.6/22.7 *	92.6	51.4/49.1	32.3/50.0	G3: 45.4/44.5	12.8/8.6	7.1/10.3	NR	NR	19.1/21.3
Yoo, 2017 [26]	63.8/63.9 ^b^	75.4/73.0	18.8/14.2	100.0	60.9/50.3	24.6/29.8	HG: 47.1/49.8	17.4/13.5	4.3/7.2	0/0	Excluded	Excluded

#—data are presented for patients who underwent URS regardless of following RNU time; ##—data are presented for patients who underwent URS regardless of biopsy status; ^a^—*median*; ^b^—*mean*; *—statistically significant difference between URS (+) and URS (−) groups. Abbreviations: AC = adjuvant chemotherapy; BC = bladder cancer; CIS = carcinoma in situ; CTX = chemotherapy; HG = high grade; LG = low grade; LNI = lymph node invasion; NA = not applicable; NAC = neoadjuvant chemotherapy; NR = not reported; RNU = radical nephroureterectomy; URS = ureteroscopy.3.3. Meta-Analysis Results.

**Table 3 jcm-10-04197-t003:** Results of exploratory subgroup analyses comparing intravesical recurrence-free survival between URS (+) and URS (−) groups, stratified by: biopsy status, URS technique, tumour location, bladder cancer history, bladder cuff excision, receipt of perioperative systemic chemotherapy, receipt of intravesical installation post RNU, and geographical region.

Stratification Variable	Subgroup	No. of Studies, *n* [Reference]	No. of Patients, *n* URS (+)/URS (−)	HR (95% CI) URS (+) vs. URS (−)	*p*-Value	Heterogeneity, I^2^ (%)	Model
Biopsy status	Performed	5 [17,21,24,25,26]	1088/1062	1.38 (1.20–1.60)	<0.001	0%	FE
	Not performed	2 [24,25]	146/395	1.28 (0.90–1.80)	0.16	0%	FE
	Other *	8 [11,13,15,16,18,19,23]	860/1453	1.57 (1.34–1.86)	<0.001	45%	FE
URS technique	Rigid	1 [20]	115/281	1.44 (1.00–2.08)	0.05	NA	NA
	Flexible	1 [11]	70/23	4.04 (1.37–11.9)	0.01	NA	NA
	Other *	11 [13,15,16,17,18,19,20,21,23,24,25,26]	1909/2374	1.42 (1.27–1.60)	<0.001	37%	FE
Tumour location	Pelvis	1 [26]	27/158	2.06 (1.16–3.64)	0.01	NA	NA
	Ureter	1 [26]	42/160	1.02 (0.62–1.67)	0.95	NA	NA
	Other *	12 [11,13,15,16,17,18,19,20,21,23,24,25]	2025/2360	1.46 (1.30–1.64)	<0.001	41%	FE
Bladder cancer history	Yes	1 [17]	61/87	0.93 (0.65–1.33)	0.69	NA	NA
	No	10 [11,13,16,17,18,19,20,23,24,25]	1553/1861	1.71 (1.48–1.97)	<0.001	18%	FE
	Other *	3 [15,20,26]	239/752	1.26 (1.00–1.57)	0.05	0%	FE
Bladder cuff excision	Performed	9 [11,13,15,16,17,19,20,25,26]	1199/2328	1.43 (1.26–1.63)	<0.001	47%	FE
	Not performed	NA	NA	NA	NA	NA	NA
	Other *	4 [18,21,23,24]	895/260	1.47 (1.16–1.85)	0.001	26%	FE
Perioperative systemic CTX	Yes	NA	NA	NA	NA	NA	NA
	No	6 [11,15,18,19,20,26]	464/1388	1.41 (1.18–1.69)	<0.001	31%	FE
	Other *	7 [13,16,17,21,23,24,25]	1630/1290	1.46 (1.27–1.68)	<0.001	45%	FE
Intravesical installation post RNU	Yes	NA	NA	NA	NA	NA	NA
	No	4 [16,17,20,26]	485/994	1.48 (1.04–2.09)	0.03	63%	RE
	Other *	9 [11,13,15,18,19,21,23,24,25]	1609/1654	1.47 (1.29–1.68)	<0.001	26%	FE
Geographical region	Asia	9 [13,15,17,18,19,20,21,25,26]	1218/2289	1.38 (1.21–1.56)	<0.001	0%	FE
	North America	2 [23,24]	711/267	1.46 (1.14–1.86)	0.002	45%	FE
	Europe	2 [11,16]	165/122	3.66 (2.01–6.64)	<0.001	0%	FE

* Heterogeneous population in terms of stratification variable or data not reported. Abbreviations: CTX = chemotherapy; FE = fixed effect; NA = not applicable; RNU = radical nephroureterectomy; URS = ureteroscopy; RE = random effect.

## Data Availability

Not applicable.

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
