# Peer review of "The Impact of Diagnostic Ureteroscopy Prior to Radical Nephroureterectomy on Oncological Outcomes in Patients with Upper Tract Urothelial Carcinoma: A Comprehensive Systematic Review and Meta-Analysis"

_jcm, 2021, doi:10.3390/jcm10184197_

Round 1
Reviewer 1 Report
The authors should review the September 2021 publication by Vidit S et al in the Journal of Urology on the same subject and include in the paper. The submitted article is too long. The methods section does not need to include the definition of PICOS. The methods should merely allow the reader to understand how the data was generated and, if necessary, to reproduce the experiment. The table of all the articles is terribly large and am really not sure how it would even be reproduced. As a metaanalysis paper, figures 2 and 3 are the requisite images that describe the veracity of the dataset I would rather see the Hazard Ratio plots much larger as those represent the entire purpose of the study
Author Response
In reference to the decision of major revisions for the jcm-1322133 manuscript, we are submitting a revised version of the article. All issues raised by the reviewers have been meticulously corrected. A detailed report on the amendments is presented below (all changes are additionally highlighted in the manuscript). Our manuscript has undergone linguistic proofreading. However, if the Reviewers still request additional language corrections, we will immediately send our manuscript to MDPI English Editing Service.
Reviewer #1:
- “The authors should review the September 2021 publication by Vidit S et al in the Journal of Urology on the same subject and include in the paper.”
Our response: A study conducted by Sharma et al. has already been included in our systematic review and meta-analysis (please see reference 24). Our latest literature search was performed on 30 June 2021 and this publication had “online ahead of print” status at that time. As paper by Sharma et al. was finally published in the September issue of the Journal of Urology, we changed the citation and removed “[published online ahead of print 2021 Apr 28]” note (References, line 499 - 501).
- “The methods section does not need to include the definition of PICOS. The methods should merely allow the reader to understand how the data was generated and, if necessary, to reproduce the experiment.”
Our response: The definition of PICOS was removed as this acronym is commonly used in evidence-based practice (Chapter 2.2, line: 102 - 107). The whole “Materials and Methods” section was reviewed again and all paragraphs were shortened to make this section clearer and easier to read.
- “The table of all the articles is terribly large and am really not sure how it would even be reproduced.”
Our response: We agree with Reviewer that table including clinical and pathological characteristics of eligible studies is large and therefore it would be hard to reproduce it. Also, it could be tiresome for the readers to analyse it. For this reason, we modified it and changed the template. We hope that all extracted data are now presented in more accessible form (Chapter 3.2, line 213 - Table 2).
- “As a metaanalysis paper, figures 2 and 3 are the requisite images that describe the veracity of the dataset I would rather see the Hazard Ratio plots much larger as those represent the entire purpose of the study”
Our response: We totally agree with Reviewer that forest plots are one of the most important components of the meta-analysis. Figure 2. was modified and divided into separate parts for each oncological outcome (Chapter 3.3: line 235 - Figure 3, line 243 - Figure 4, line 251 - Figure 5, line 259 - Figure 6. Additionally, results of risk of bias evaluation were presented in a separate Figure (Chapter 3.1, line 176 - Figure 2). We hope these changes allowed for better visualization of forest charts with all necessary data.
Reviewer 2 Report
Dear editor,
thank you for giving me the opportunity to review to manuscript titled “The impact of diagnostic ureteroscopy prior to radical nephroureterectomy on oncological outcomes in patients with upper tract urothelial carcinoma: a comprehensive systematic review and meta-analysis”; in which the authors if ureteroscopy can increase the rate of intravesical recurrence of urothelial carcinoma after nephroureterectomy for upper tract urothelial carcinoma (UTUC).
Abstract
The abstract is well written and gives a good overview of the topic, the results and the conclusion. The authors state that “ A systematic literature 38 search of the PubMed, Embase and Cochrane Library databases was performed in June 2020 (page 1 line 39). I assume they mean 2021, as in the rest of manuscript?
Introduction
The introduction is well written and gives a good overview over the topic. However, current treatment strategies such as a kidney sparing approach in low-risk disease or intravesical installation of chemotherapy should be discussed more in depth (see PMID 33052841 for example).
Methods
The methods are fully explained. Was the review registered at Prospero?
Did the authors think about collecting information with respect to urine cytology (e.g., barbotage cytology)?
Results
Chapter 3.1. should all be explained along site the figure (as a caption) and not within the manuscript, this makes the manuscript unnecessarily long.
There is a typo in chapter 3.2. page 5 line 194: What amount and what percentages?
Can the authors calculate a pooled-follow up time?
“Also, no association between URS and worse IVRFS was 310 found in the subset of patients with history of bladder tumor” -> please the hazard ratio and 95% CI and p-value for this statement (from table 3; makes is easier to read)
Discussion
There is a typo page 18 line 325; Kim et al.
The authors should try to use they’re in own findings and formulate clear and precise recommendation with respect to the treatment algorithm. Should all patients undergo URS and biopsy, despite the higher risk of bladder tumor recurrence? Which patients should not undergo biopsy? The authors should try to discuss their results and put them into a clinical perspective. They should also discuss the role of urine cytology in this context.
Concern:
A major concern is the lack of data with respect to the effect of intravesical installation post RNU. This is recommended by the guidelines has a significant impact on the rate of intravesical recurrence.
Other than that, the author present a couple of clinical relevant results. However, they should try to put their findings into the clinical context and try to give precise recommendation along with their results.
Author Response
In reference to the decision of major revisions for the jcm-1322133 manuscript, we are submitting a revised version of the article. All issues raised by the reviewers have been meticulously corrected. A detailed report on the amendments is presented below (all changes are additionally highlighted in the manuscript). Our manuscript has undergone linguistic proofreading. However, if the Reviewers still request additional language corrections, we will immediately send our manuscript to MDPI English Editing Service.
- “The abstract is well written and gives a good overview of the topic, the results and the conclusion. The authors state that A systematic literature 38 search of the PubMed, Embase and Cochrane Library databases was performed in June 2020 (page 1 line 39). I assume they mean 2021, as in the rest of manuscript?”
Our response: It was a typo and appropriate correction was made (Abstract, line 39)
2.”The introduction is well written and gives a good overview over the topic. However, current treatment strategies such as a kidney sparing approach in low-risk disease or intravesical installation of chemotherapy should be discussed more in depth.”
Our response: As suggested, kidney sparing approach was additionally discussed in the introduction section (Chapter 1, line 62 - 78). To avoid making introduction unnecessarily long, we discussed additional issues regarding intravesical installation of chemotherapy in the discussion section (Chapter 4, line 361 - 376).
- “The methods are fully explained. Was the review registered at Prospero?”
Our response: Our study has been registered at Prospero, however, we are still waiting for assigned CRD number. Based on our experience with previous systematic reviews and meta-analyses, registration time could reach even 3 - 6 months. We did not want to delay submission of this study and decided to submit our manuscript without CRD number. However, we hope that CRD number for this study will be assigned soon and we can provide it in the manuscript .
- “Did the authors think about collecting information with respect to urine cytology (e.g., barbotage cytology)?”
Our response: We planned to collect and present data regarding urine cytology. Unfortunately, there is a paucity of data regarding this issue in included studies. We added appropriate information in Chapter 3.2 (line 196 -198) and Chapter 4 (line 341 - 345), as well as mentioned it in the paragraph regarding limitations of the following study (Chapter 4, line 402 - 405).
- “Chapter 3.1. should all be explained along site the figure (as a caption) and not within the manuscript, this makes the manuscript unnecessarily long.”
Our response: As suggested, Chapter 3.1 was explained along site the figure as a caption. We also combined Chapter 3.1 with Chapter 3.2.
- “There is a typo in chapter 3.2. page 5 line 194: What amount and what percentages?”
Our response: Appropriate correction of this sentence was made (Chapter 3.1, line 152 - 154)
- “Can the authors calculate a pooled-follow up time.”
Our response: A pooled follow-up time was calculated and appropriate sentence was added in the Chapter 3.1 (line 158)
- “Also, no association between URS and worse IVRFS was 310 found in the subset of patients with history of bladder tumor” -> please the hazard ratio and 95% CI and p-value for this statement (from table 3; makes is easier to read).”
Our response: The hazard ratio, 95% CI and p-value was added for this statement (Chapter 3.3.2, line 280 - 281)
- “There is a typo page 18 line 325; Kim et al.”
Our response: Appropriate correction of this sentence was made (Chapter 4, line 294).
- “The authors should try to use they’re in own findings and formulate clear and precise recommendation with respect to the treatment algorithm. Should all patients undergo URS and biopsy, despite the higher risk of bladder tumor recurrence? Which patients should not undergo biopsy? The authors should try to discuss their results and put them into a clinical perspective. They should also discuss the role of urine cytology in this context.”
Our response: We included concise recommendations in the discussion section (Chapter 4, line 383 - 391).
Round 2
Reviewer 1 Report
Line 77. “…it allows to detect…” should be “CTU may detect a significant…”
Line 98 should have “a” preceding “combination”
Quotation marks follow the European custom of lower and then upper marks rather than two upper marks.
In Section 2.2, authors should include in parentheses the word represented by PICO
Line 152 should be shortened from “Throughout the whole process of methodological quality…” to “During the study…”
Line 157 Authors should include the word “in” preceding “…the Cochrane”
Line 275 Authors should include the word “the” preceding “majority”
The large figure on page 5 depicts both forest plots and bias dot diagrams but is not numbered (it is in the sequence between figures 6 and 7) and the legend is crossed out. That figure appears pixelated and will likely need better quality. Can the risk of bias diagram data allow figure 2 to be deleted?
Line 386 probably doesn’t need “updated”
Line 427 should be “consistent” not “consistently”
The paragraph in the discussion starting with 492 would suggest that ureteroscopic biopsy may not be indicated prior to nephro-ureterectomy. Instead, the authors may wish to the results of the POUT trial and the potential survival advantage of neoadjuvant platinum-based chemotherapy (NAC). As most oncologists would not begin NAC without biopsy, ureteroscopic biopsy may continue to play a role even in cases in which pre-operative imaging and cytology is highly suggestive of UC. Upfront RNU, as the authors suggest, without biopsy would exclude such patients from the potential benefit of NAC.
Author Response
In reference to the decision of minor revisions for the jcm-1322133 manuscript, we are submitting a revised version of the article. All issues raised by the Reviewer have been corrected. A detailed report on the amendments is presented below (all changes are additionally highlighted in the manuscript).
Reviewer #1:
- Line 77. “…it allows to detect…” should be “CTU may detect a significant…”.
Our response: Sentence was corrected (Chapter 3.1, line 152 - 154).
- Line 98 should have “a” preceding “combination”.
Our response: Sentence was corrected (Chapter 2.1, line 92 - 93).
- Quotation marks follow the European custom of lower and then upper marks rather than two upper marks.
Our response: Quotation marks were corrected (Chapter 2.1, line 93 - 96).
- In Section 2.2, authors should include in parentheses the word represented by PICO.
Our response: The words represented by PICOS acronym were added in parentheses (Chapter 2.2, line 102 - 103).
- Line 152 should be shortened from “Throughout the whole process of methodological quality…” to “During the study…”.
Our response: Sentence was shortened and moved to the end of the paragraph (Chapter 2.4, line 133 - 134).
- Line 157 Authors should include the word “in” preceding “…the Cochrane”.
Our response: Sentence was corrected (Chapter 2.4, line 128 - 129).
- Line 275 Authors should include the word “the” preceding “majority”.
Our response: Sentence was corrected (Chapter 3.2, line 198, 217).
- The large figure on page 5 depicts both forest plots and bias dot diagrams but is not numbered (it is in the sequence between figures 6 and 7) and the legend is crossed out. That figure appears pixelated and will likely need better quality. Can the risk of bias diagram data allow figure 2 to be deleted?
Our response: We improved the quality of the Figure 7. We are not completely sure how to understand a comment regarding numeration and legend. Each funnel plot in Figure 7 has a letter (A,B,C,D), which is described in the figure legend (A - intravesical recurrence-free survival, B - cancer specific survival, C - overall survival, D - metastases-free survival). As the evaluation of both publication bias (presented in Figure 7) and confounding bias (presented in Figure 2) are an essential parts of systematic review and meta-analysis (according to PRISMA statement and Cochrane Handbook for Systematic Reviews of Interventions), we think it is reasonable to keep both figures.
- Line 386 probably doesn’t need “updated”.
Our response: Sentence was corrected (Chapter 4, line 288).
- Line 427 should be “consistent” not “consistently”.
Our response: Sentence was corrected (Chapter 4, line 329).
- The paragraph in the discussion starting with 492 would suggest that ureteroscopic biopsy may not be indicated prior to nephro-ureterectomy. Instead, the authors may wish to the results of the POUT trial and the potential survival advantage of neoadjuvant platinum-based chemotherapy (NAC). As most oncologists would not begin NAC without biopsy, ureteroscopic biopsy may continue to play a role even in cases in which pre-operative imaging and cytology is highly suggestive of UC. Upfront RNU, as the authors suggest, without biopsy would exclude such patients from the potential benefit of NAC.
Our response: We fully agree with Reviewer that URS biopsy (or e.g. percutaneous biopsy) is indicated when NAC is planned. In the discussion section, we included information regarding beneficial effect of NAC and AC, based on the results of recent meta-analysis conducted by Leow et al. (including results of POUT trial) (Chapter 4, line 378 - 379). Also, in the summary paragraph we included additional statement regarding the role of URS biopsy in NAC setting (Chapter 4, line 386 - 394).